# Impact of Experimental Conditions on Extracellular Vesicles’ Proteome: A Comparative Study

**DOI:** 10.3390/life13010206

**Published:** 2023-01-11

**Authors:** Tímea Böröczky, Gabriella Dobra, Mátyás Bukva, Edina Gyukity-Sebestyén, Éva Hunyadi-Gulyás, Zsuzsanna Darula, Péter Horváth, Krisztina Buzás, Mária Harmati

**Affiliations:** 1Laboratory of Microscopic Image Analysis and Machine Learning, Institute of Biochemistry, Biological Research Centre, Eötvös Lorand Research Network (ELKH), H-6726 Szeged, Hungary; 2Department of Immunology, Albert Szent-Györgyi Medical School, Faculty of Science and Informatics, University of Szeged, H-6720 Szeged, Hungary; 3Doctoral School of Interdisciplinary Medicine, Albert Szent-Györgyi Medical School, University of Szeged, H-6720 Szeged, Hungary; 4Laboratory of Proteomics Research, Biological Research Centre, Eötvös Lorand Research Network (ELKH), H-6726 Szeged, Hungary; 5Single Cell Omics Advanced Core Facility, Hungarian Centre of Excellence for Molecular Medicine, H-6726 Szeged, Hungary; 6Institute for Molecular Medicine Finland (FIMM), University of Helsinki, 00290 Helsinki, Finland

**Keywords:** small extracellular vesicles (sEVs), sEV proteome, mass spectrometry, nanoparticle tracking analysis (NTA), pathway enrichment analysis, EV-depleted fetal bovine serum (FBS), serum-free medium, FBS starvation

## Abstract

Extracellular vesicle (EV) research is a rapidly developing field, mainly due to the key role of EVs in intercellular communication and pathophysiological processes. However, the heterogeneity of EVs challenges their exploration and the establishment of gold-standard methods. Here, we aimed to reveal the influence of technical changes on EV biology and the reliability of experimental data. We used B16F1 melanoma cells as a model and applied nanoparticle tracking analysis, mass spectrometry (LC-MS/MS) and pathway enrichment analysis to analyze the quantity, size distribution, proteome and function of their small EVs (sEVs) produced in sEV-depleted fetal bovine serum (FBS)-containing medium or serum-free medium. Additionally, we investigated the effects of minor technical variances on the quality of sEV preparations. We found that storage of the isolates at −80 °C has no adverse effect on LC-MS/MS analysis, and an additional washing step after differential ultracentrifugation has a minor influence on the sEV proteome. In contrast, FBS starvation affects the production and proteome of sEVs; moreover, these vesicles may have a greater impact on protein metabolism, but a smaller impact on cell adhesion and membrane raft assembly, than the control sEVs. As we demonstrated that FBS starvation has a strong influence on sEV biology, applying serum-free conditions might be considered in in vitro sEV studies.

## 1. Introduction

Extracellular vesicle (EV) research is a rapidly developing field, mainly due to the key role of EVs in intercellular communication. Their nanoscale size, lipid-bilayer-enclosed specific molecular cargo and targeted delivery make them excellent information-shuttle vehicles. As they are released by all mammalian cells, enter the circulation and create a body-wide communication network, EVs have an essential role in a broad range of physiological and pathological processes [1,2]. In the context of tumor biology—which is one of the most studied fields of EV research—they participate in all stages of the disease, supporting cell proliferation, cell migration, inflammatory responses, immune suppression, immune evasion, epithelial-to-mesenchymal transition, angiogenesis, metastasis and therapy resistance, as well [3,4,5]. EVs originate either from the endosomal system (i.e., exosomes) or from the plasma membrane (i.e., microvesicles/ectosomes) and they comprise diverse subpopulations that can also differ in size, morphology, composition or biological activity [1]; this requires specialized experimental techniques to study and complicates data comparisons from different resources [6,7,8,9]. 

This heterogeneity greatly challenges their exploration, not just in in vivo studies, but in more controllable in vitro experiments, as well; therefore, there are currently no gold-standard techniques for EV characterization, separation or functional studies [10].

To increase and maintain rigor and reproducibility in EV research, the International Society for Extracellular Vesicles (ISEV) have made several efforts; for instance, they proposed Minimal Information for Studies of Extracellular Vesicles (MISEV) guidelines [11,12] and established the Rigor & Standardization Subcommittee and Task Force Initiatives. They recommend choosing methodologies that best fit the research question, but they suggest a comprehensive report of the applied conditions and technologies. However, without a clear insight into the influence of the technical parameters—such as culture conditions, isolation protocols or storage—on EV biology and the reliability of experimental data, it can be difficult to set up the most appropriate experimental protocols. Thus, a number of publications still address this issue [13,14], and more of these are required.

The first technical parameter which can affect the experimental data of in vitro EV studies is the cell culturing condition, including the serum supplementation of culture media. As serum is rich in EVs, RNA and protein aggregates, it can contaminate the EV isolates, which might influence the subsequent analyses and functional studies and lead to misinterpretations. Furthermore, serum products are complex mixtures with undefined composition that can vary drastically among different batches depending on the manufacturer and lot number [15,16,17,18]. To overcome these issues, EV-depleted serum or serum-free medium is used in most in vitro EV studies. However, these changes in the culture media can affect its capability to maintain cell growth; they can modify the cell phenotype or lead to stress responses, which may influence the quantity and content of the released EVs [17]. The second decisive technical step in EV studies is the isolation/concentration procedure. Today, there are a series of isolation methods based on EV size, density, marker expressions or their combinations. However, they result in quite different EV purities and yields [19,20]. Additionally, the storage conditions of EVs can also affect downstream analyses and functional studies [21].

Here, to reveal previously unexplored aspects, we aimed to use a sensitive method which can reflect the effects of different technical variables on vesicle production, cargo and the purity of EV preparations. We applied mass spectrometry and analyzed the fetal bovine serum (FBS) starvation-induced changes in EV biology. Additionally, we investigated the effects of some less-studied minor technical changes in the quality of EV preparations. As tumor cell-derived small EVs (sEVs) are one of the most studied vesicles, we chose the widely used B16F1 mouse melanoma cell line and its sEVs as a model.

## 2. Materials and Methods

### 2.1. Cell Culture

In this study, we used the B16F1 mouse melanoma cell line, which was obtained from ECACC (Salisbury, UK) and cultured in Dulbecco’s modified Eagle’s medium (DMEM, Lonza, Basel, Switzerland) supplemented with 10% FBS (EuroClone, Pero, Italy; or Biowest, Nuaillé, France), 2 mM L-glutamine, 1% Penicillin–Streptomycin–Amphotericin B mixture (P/S/A), 1% MEM non-essential amino acids, 1% MEM vitamin solution and 0.01% sodium pyruvate (all from Lonza, Basel, Switzerland). The B16F1 cell culture was maintained in a humidified incubator at 37 °C and 5% CO_2_. During the experiments, the morphology of the cells was monitored daily and the cells were stained with trypan blue dye (Corning, NY, USA) to count live and dead cells in a Bürker chamber to monitor cell viability. To compare different cultures, the same number of cells were seeded. The phase contrast microscopy images were made using an Axiovert S100 microscope (Zeiss, Oberkochen, Germany) at 20× magnification.

### 2.2. sEV Generation Methods

For the comparative experiments, the same number of cells were seeded and cultures were incubated in different media: (i) fresh sEV-depleted OptiClone FBS-containing medium, (ii) fresh sEV-depleted Biowest FBS-containing medium and (iii) fresh FBS-free medium. For sEV isolation, supernatants were harvested after 72 h of incubation. OptiClone and Biowest FBS were purchased from EuroClone (ECS0183L, Pero, Italy) and Biowest (S182P-500, Nuaillé, France), respectively.

For bovine EV depletion, FBS was subjected to one hour of centrifugation at 150,000× *g* (T-1270 fixed-angle rotor, Sorvall, Waltham, MA, USA), and then, the supernatant was filtered through a 0.22 µm pore-size membrane. The aliquots were stored at −20 °C until use.

### 2.3. sEV Isolation

Vesicles were isolated via filtration and differential ultracentrifugation as described previously [22,23]. Briefly, 72  h supernatants were harvested and supplemented with complete protease inhibitor cocktail (Roche, Mannheim, Germany) and centrifuged at 780× *g* for 5 min and at 4000× *g* for 15 min at 4 °C; then, filtered through a 0.22 µm pore-size membrane to remove debris, cells and larger vesicles. Small EVs were pelleted via ultracentrifugation at 150,000× *g* for 60 min at 4 °C using a T-1270 fixed-angle rotor and a WX+ ultracentrifuge (Sorvall, Thermo Fisher Scientific, Waltham, MA, USA). 

The pellet was handled differently depending on the downstream analyses. The pellet was rinsed twice and resuspended in 90 µL NP40 lysis buffer (Invitrogen, Thermo Fisher Scientific, Waltham, MA USA) for proteomic analysis or in 120 µL particle-free phosphate-buffered saline (PBS, Lonza, Basel, Switzerland) for nanoparticle tracking analysis (NTA). Three sample types were measured immediately (Fresh sEV, Ctrl sEV and w/o FBS sEV); the w/ wash sEV sample was generated via an additional washing step, i.e., re-suspension in PBS and ultracentrifugation at 150,000× *g* for 60 min at 4 °C; and the Frozen sEV sample was stored at −80 °C for 3 months (Table 1). The experimental workflow of the whole study can be seen in Appendix A.

To check the morphology of the isolated sEVs, we performed transmission electron microscopy (TEM) on the Ctrl sEVs as described previously [22]. Briefly, we used a Tecnai G2 20 X-Twin instrument (FEI, Hillsboro, OR, USA), operating at an acceleration voltage of 200 kV; the sEVs were dropped on a grid (carbon film with 200-mesh copper grids; CF200-Cu, Electron Microscopy Sciences, Hatfield, PA, USA) and dried before measurement.

### 2.4. Nanoparticle Tracking Analysis (NTA)

The concentration and size distribution of the sEV samples were measured right away (Fresh sEV, Ctrl sEV, w/o FBS sEV and w/ wash sEV) or after being stored at −80 °C (Frozen sEV) using a NanoSight NS300 instrument with 532 nm laser (Malvern Panalytical Ltd., Malvern, UK) coupled with NTA software 3.4 (Malvern Panalytical Ltd.). The samples were diluted in particle-free PBS before measurement. The following settings were used in all of the measurements: syringe pump flow rate: 30 units; temperature: 24 °C; camera level: 16; capture duration: 60 s/video; number of captures: 3; screen gain: 1; and detection threshold: 4. 

### 2.5. Proteomic Analysis by Liquid Chromatography–Tandem Mass Spectrometry (LC-MS/MS)

#### 2.5.1. Separation and In-Gel Digestion of sEV Proteins

The vesicular pellets were resuspended in an NP40 lysis buffer (Thermo Scientific) and incubated for 30 min on ice. A Pierce BCA Protein Assay Kit (Thermo Scientific) was used to measure the protein concentration of sEV isolates; then, 25 μg protein of each sample was diluted in a mixture of 4 × NuPAGE LDS Sample Buffer (Invitrogen) and 10 × NuPAGE Sample Reducing Agent (Invitrogen), boiled at 96 °C for 10 min, cooled on ice and separated in 4–12% Bis-Tris Protein Gel (NuPAGE Novex) using an Xcell SureLock Mini-Cell (Invitrogen) at 200 V and 0.03 A for 40 min with an electrophoresis buffer (NuPAGE MOPS SDS Running Buffer). All reagents were purchased from Thermo Fisher Scientific (Waltham, MA, USA). Then, the gel was stained overnight with Coomassie blue (0.1% Coomassie Brilliant Blue R-250 in 45% ethanol, 45% MQ water and 10% acetic acid), destained with a mixture of 10% acetic acid, 30% ethanol and 60% MQ water, and rinsed in MQ water. After dividing each lane into 12 equal bands, in-gel digestion was performed. Gel bands were cut into smaller segments, and the dye and the SDS were removed by washing them with 3 × 50 µL 25 mM ammonium-bicarbonate (ABC)/50% acetonitrile (AcN). After reduction with DTT (1,4-dithiothreitol, Sigma-Aldrich, St. Louis, MO, USA; 20 µL, 10 mM DTT in 25 mM ABC) at 56 °C for 30 min, and alkylation with IAM (iodoacetamide, Sigma-Aldrich, St. Louis, MO, USA; 20 µL, 55 mM IAM in 25 mM ABC) for 30 min at RT in the dark, the gel samples were dried in a vacuum centrifuge and, after that, rehydrated in 20 µL of trypsin (Sequencing Grade Modified Trypsin, Promega, Madison, WI, USA; 5 ng/µL in 25 mM ABC) and incubated at 37 °C. The addition of 2 µL of 10% formic acid (FA) to lower the pH of the buffer below 3 stopped the digestion after 4 h. Tryptic peptides were extracted from the gel with 3 × 50 µL of 2% FA in 50% AcN and dried. Prior to mass spectrometric analysis, all samples were redissolved in 50 µL of 0.1% FA.

#### 2.5.2. LC-MS/MS Analysis and Protein Identification

The samples were analyzed using an LTQ Orbitrap Elite (Thermo Fisher Scientific Waltham, MA, USA) mass spectrometer on-line coupled with a nanoHPLC (nanoAcquity, Waters, Milford, MA, USA) system. From the in-gel digested stripes, 5 µL was loaded (for 3 min at 8 µL/min flow, using 0.1% FA in 3% AcN-97% water) onto a reversed-phase trap column (Waters, Symmetry C18, 0.18 × 20 mm) and separated using a C18 reversed-phase nanocolumn (Waters, BEH300C18 1.7 µm) (0.075 × 200 mm). The flow rate was 330 nL/min and a linear gradient was used of 3% to 40% solvent B in 37 min (solvent A was 0.1% FA in water and solvent B was 0.1% FA in AcN). Through the liquid junction between the chromatographic column and the non-coated silica nanospray emitter (NewObjective, Woburn, MA, USA; 10 µm tip ID), a high voltage of 1.2 kV was applied. The mass spectrometer operated in data-dependent mode: the survey mass spectra were detected in the Orbitrap with a high resolution (R = 60 k at *m*/*z*: 400, mass range *m*/*z*: 380–1400) and the most abundant multiply charged 20 peaks were selected for ion-trap fragmentation (NCE: 35%; activation q: 0.25; activation time: 10 ms; and minimum signal intensity: 5000 counts). The MS/MS spectra were detected in the ion trap. Dynamic exclusion was used, and the precursors were excluded for 15 s after the first fragmentation event. 

The database search and quantitative analysis were performed using Proteome Discoverer 2.4 software (Thermo Fisher Scientific, Waltham, MA, USA). The Sequest HT search engine was used with the following parameters: parent ion tolerance: 5 ppm; fragment ion tolerance: 0.6 Da; Cys carbamidomethylation was set as constant and Met oxidation, cyclisation of peptide N-terminal Glu to pyroglutamic acid, and protein N terminal acetylation were set as variable modifications. Only fully tryptic peptides were considered with a maximum of 2 missed cleavage sites. Only proteins identified with high confidence (*q* < 0.001) and with at least 3 unique peptides were accepted (Appendix A shows the raw proteomic data). 

### 2.6. Data Analysis

Welch’s test was used to compare the means of continuous variables (sEV mean size and concentration, and sEV/cell) between two independent groups without checking the assumptions of normal distribution and homoskedasticity [24]. For more than two groups, we used the Shapiro–Wilk test to check the normality and the Brown–Forsythe test to determine the homogeneity of variance. Then, we applied one-way ANOVA with Tukey’s post hoc test on variables with normal distribution (cell number/culture, cell viability, sEV concentration and sEV mean size), and Welch’s ANOVA with Games–Howell post hoc test in case of heteroskedasticity (intensity of mouse and bovine proteins).

To reduce standard deviation and skewness, some variables were logarithmized in base two.

Pearson’s method was used to test the correlation between continuous variables.

For the statistical analyses, GraphPad Prism 8.4.3 (San Diego, CA, USA) was used. A value of *p* < 0.05 was considered significant in each statistical test. 

For hierarchical clustering, row centering and unit variance scaling was applied to the protein intensities. On the heatmap, both rows and columns are clustered using 1-Pearson’s correlation distance and complete linkage. The clustering procedure was carried out using the MORPHEUS software (Morpheus by Broad Institute RRID:SCR_017386, https://software.broadinstitute.org/morpheus; last accessed on 12 December 2022).

Pathway enrichment analyses were performed using ShinyGO 0.76.3 with the default parameters. As a measure of effect size, fold enrichment indicates how the proteins of certain pathways are overrepresented. Fold Enrichment is defined as the percentage of proteins in our data belonging to a pathway, divided by the corresponding percentage in the ShinyGO background [25].

## 3. Results

### 3.1. Culture Medium Affected the Morphology and Proliferation, but Not the Viability of the Cells

In this study, we used the B16F1 mouse melanoma cell line as a model. Two different serum supplementations, i.e., 10% sEV-depleted Opticlone or Biowest FBS and serum-free culture medium were applied to compare their impact on cell growth, viability and morphology. We observed that cell morphology did not differ between the two FBS-containing media; in contrast, the cells were elongated in the serum-free cultures (Figure 1A). 

Moreover, the cell cultures in the sEV-depleted Biowest-supplemented medium had a significantly higher cell number compared to the sEV-depleted OptiClone-supplemented medium (*p* = 0.0413) and the serum-free medium (*p* = 0.0033). Cell viability slightly decreased in the serum-free medium, but the differences between the three cultures were not significant (Figure 1B).

### 3.2. FBS Starvation of Donor Cells Influenced the Concentration and the Size of sEVs

The isolation of B16F1 melanoma cell-derived sEVs was optimized in previous studies [22,23], where the vesicles were widely characterized using (i) a high-resolution imaging technique, such as atomic force microscopy (AFM), (ii) a single-particle analysis technique that estimates light scattering properties, such as dynamic light scattering (DLS), and (iii) Western blot analysis (WB), as well. Here, we performed transmission electron microscopy (TEM) for quality control of the isolates (Appendix A).

In this study, we used five different sample types to examine the influencing effect of the culturing, isolation and storage conditions on the donor cells and cell-derived sEVs (Table 1). The experimental workflow is presented on Appendix A.

First of all, sEV samples were measured via nanoparticle tracking analysis (NTA), which showed that vesicles isolated from all three differently conditioned media had a size range of 107.5–111.5 nm. The mean size of the vesicles did not differ after an additional washing stage or storage at −80 °C (Figure 2A). The concentration, mean size values and distribution curves are presented in Appendix A.

Freezing had no effect on the sEV concentration of isolates, but the lack of serum did. A lower concentration of particles was measured in the sEV samples isolated from the w/o FBS supernatant sample compared to the Ctrl sEV and the w/ wash sEV samples (*p* = 0.0069, *p* = 0.0355) (Figure 2B). However, as the cell proliferation considerably decreased, the number of vesicles released per cell was significantly higher in the w/o FBS sample (*p* < 0.0001) than in the other ones (Figure 2C).

### 3.3. Technical Parameters of In Vitro sEV Studies May Affect the Vesicular Proteomic Data

#### 3.3.1. sEV Storage at −80 °C Had No Adverse Effect on LC-MS/MS-Based Protein Identification

After cells were cultured in sEV-depleted FBS-containing medium, the supernatants were collected. Following differential centrifugation, the sEV samples were subjected to LC-MS/MS immediately, without freezing (referred to as Fresh sEV), or later, after storage, at −80 °C (referred to as Frozen sEV). 

In the first step, we analyzed the overlap between the two sEV samples’ protein lists to determine if the freezing influenced the efficiency of mass spectrometry-based protein identification. We found that 99.30% (428 of 431) of the proteins overlapped in the Fresh sEV and Frozen sEV samples (Figure 3A), suggesting that sEV storage at −80 °C did not reduce the efficiency of proteomic analysis.

In the second step, Pearson’s correlation analysis was used to compare the identified proteomic datasets of the two sEV samples (Appendix A). There were no significant differences in the proteomic profile of the freshly measured or frozen sEV isolates. The correlation matrix demonstrates the high degree of similarity between the two samples (*r* = 0.95) (Figure 3B), suggesting that the storage of sEV isolates at −80 °C had no adverse effect on LC-MS/MS-based protein identification.

#### 3.3.2. FBS Starvation of Donor Cells Markedly Changed the Protein Content of Vesicles

In the next proteomic analysis, we applied FBS starvation during sEV production (w/o FBS sEVs) or an additional washing step (w/ wash sEVs) during sEV isolation to determine their influencing effects on the sEV proteome and isolation efficiency, respectively. The samples produced in sEV-depleted FBS-containing medium were used as a control (Ctrl sEV).

Comparison of the entire protein lists of the three sample types revealed that starvation markedly influenced the protein composition of the vesicles. We found that 11.33% (103 of 909) of the identified proteins were exclusively detected in the w/o FBS sEV isolate and 9.68% (88 of 909) of proteins were missing from this sample. The analysis also revealed a 91.19% (735 of 806) overlap between the proteome of the Ctrl sEV and w/ wash sEV samples (Figure 4A). 

Pearson’s correlation analysis was performed on the protein content of the Ctrl sEV, w/o FBS sEV and w/ wash sEV samples (Appendix A). There were no significant differences in the proteomic profile of the Ctrl sEV and w/ wash sEV isolates; in contrast, the w/o FBS sEV samples dramatically differed from the two other samples. The correlation matrix demonstrates the high degree of similarity between the Ctrl sEV and w/ wash sEV isolates (*r* = 0.95), and the large differences between the w/o FBS sEV sample and the two other ones (*r* = 0.07) (Figure 4B). 

Then, to determine the ratio of FBS-derived, contaminating proteins and B16F1-derived proteins in the sEV isolates, the identified proteins were grouped based on their origin, i.e., bovine or mouse. In the analysis of only mouse proteins, the ratio of unique proteins in the w/o FBS sEV sample was of similar magnitude (14.97%, 97 of 648) to the case of the entire proteome analysis. However, this ratio decreased to 1.35% (3 of 222) in the analysis of the bovine proteins. Comparing the Ctrl sEVs and w/ wash sEVs using either the mouse or bovine proteins, we found that the mouse proteins overlapped in 87.84% (484 of 551), whereas the bovine proteins overlapped in 99.09% (217 of 219) (Figure 4C,D).

It must be noted that 4.29% of the entire proteome could have been both mouse and bovine proteins; therefore they were excluded from the Venn diagram in Figure 4C,D. Consequently 71.29% (648 of 909) of the identified proteins were derived exclusively from the cultured mouse cells and 24.42% (222 of 909) of the identified proteins were derived exclusively from the FBS supplementation.

Based on a comparison of the complete or the narrowed (i.e., mouse or bovine) protein lists of the three sample types, we can conclude that FBS starvation of the donor cells, but not the additional washing step, significantly altered the protein composition of the vesicles.

#### 3.3.3. Protein Cargo of sEVs Plays a Role in Various Biological and Molecular Processes

The protein intensity values of the Ctrl sEV, w/ wash sEV and w/o FBS sEV samples were subjected to row centering and unit-variance scaling for hierarchical clustering. The resulting heatmap demonstrates that the Ctrl sEV and w/ wash sEV samples were highly similar, although there were still some differences between them. In contrast, the log_2_ (intensity z-score) of the w/o FBS sEVs showed opposing values to the other two samples, due to its unique proteomic profile (Figure 5A). These sEVs were produced in serum-free medium, and as we expected, they showed an elevated mouse protein level (*p* < 0.0001) and a decreased bovine protein level (*p* < 0.0001) (Figure 5B,C). 

In addition to the hierarchical clustering, we performed pathway enrichment analysis both for the mouse and bovine proteins using ShinyGO 0.76.3 software to reveal the biological processes and molecular functions that can be associated with the identified sEV protein dataset of the Ctrl sEV sample. The analysis confined to mouse proteins suggested that melanoma cell-derived sEVs may contribute to protein metabolism and purine metabolism upon uptake by the sEV target cells (Figure 5D,E). Bovine protein analysis revealed that the co-isolated FBS sEVs may participate in the metabolism of carbohydrates and carboxylic acid; may regulate endopeptidase, peptidase, hydrolase, or oxidoreductase activity; and may induce small-molecule (e.g., lipid or protein) binding (Figure 5F,G).

Our data indicate that the protein cargo of sEVs plays a role in various biological and molecular processes.

#### 3.3.4. FBS Starvation May Increase the Activity of Released sEVs in Protein Metabolism and Molecular Signaling Processes in Target Cells

Then, we aimed to determine the impact of FBS starvation—during sEV production—or the additional washing step—in the isolation protocol—on the protein composition of melanoma sEVs. We used the Ctrl sEV sample—which was produced in 10% sEV-depleted FBS-containing medium—as a baseline for further pathway enrichment analyses (Figure 6).

Compared to the Ctrl sEVs, elevated mouse proteins in the w/o FBS sEV sample were predicted to be involved in protein metabolism and molecular signaling processes such as cytoplasmic translation, protein-containing complex assembly or GDP binding (Figure 6A,C). Mouse proteins, which showed decreased intensity here, may induce cell adhesion, membrane raft assembly, ion channel activities or clathrin binding (Figure 6B,D), suggesting that the Ctrl sEVs have greater potential to support these processes than the w/o FBS sEVs.

We also attempted to determine whether an additional washing step in the isolation protocol decreased the yield of sEVs and modified the ratio of the different sEV populations in the isolates. Therefore, we checked if the biological processes and molecular functions of the w/ wash sEVs were altered compared to the Ctrl sEVs (Appendix A). We found that proteins with altered intensity values may impact different processes of protein translation. 

Based on our findings, we can conclude that FBS starvation may increase the activity of released sEVs in protein metabolism and molecular signaling processes in target cells.

## 4. Discussion

The essential role of extracellular vesicles (EVs) in physiological and pathological processes has been widely demonstrated in recent decades. In spite of the great progress in their research, the heterogeneity of EVs challenges their investigation [10]. 

Since there are no gold-standard EV research methods, the Minimal Information for Studies of Extracellular Vesicles (MISEV) guidelines were published by the International Society for Extracellular Vesicles (ISEV) to enhance the reliability, transparency and reproducibility of EV studies [11,12]. However, the selection of the experimental techniques and the establishment of the protocols that best fit the research question can still be difficult without full insight into their impact on EV biology and reliability of experimental data. 

Therefore, we aimed to deeply characterize the FBS starvation-induced changes in EV biology and the effects of two minor technical parameters on the quality of EV preparations. We performed mass spectrometry-based analysis of B16F1 mouse melanoma cell line-derived sEVs, which was suitable to determine the full proteome of sEV preparations and the ratio of the melanoma-derived and contaminating FBS proteins, as well. As the EV cargo reflects the physiological state of donor cells [17,22], proteomic analysis is considered a sensitive method for monitoring the impact of culture conditions on EV biology.

Fetal bovine serum (FBS) is a common supplement for cell culture media that contaminates sEV samples with exogenous EVs and other particles, such as lipoproteins or protein complexes [15,17,18]. As an alternative, it is suggested that EV-depleted FBS be used. However, depletion protocols cannot remove EVs completely and the resulting FBS is not EV-free [17,18,26,27]. Another alternative solution is the usage of serum-free medium, but it may induce cellular stress and cause changes in the cellular phenotype, EV cargo packaging and release mechanisms [17]. In accordance with these findings, here, we demonstrated that serum-free medium has a reduced ability to support cell growth, induces cellular stress and alters the sEV release of donor cells, as the investigated B16F1 melanoma cultures showed decreased proliferation, altered morphology and increased sEV production in FBS-free medium compared to the cultures maintained in sEV-depleted FBS-containing medium. Additionally, we provided a detailed description of the sEVs’ proteomic changes upon FBS starvation; the w/o FBS sEV sample showed a decreased ratio of contaminating FBS proteins, which can help to prevent the misinterpretation of experimental data, but its unique protein profile suggested alterations in protein packaging in the donor cells. Pathway enrichment analyses predicted dramatic changes in the biological activity of their sEVs, as well. Analysis of the mouse proteins suggested that compared to the Ctrl sEVs, they have greater potential to induce protein metabolism and molecular signaling processes upon uptake by the sEV target cell, but their capability to support cell adhesion, membrane raft assembly and ion channel activities is decreased.

Wei et al. also investigated vesicular cargo changes. Using U251 glioma cells, they showed that the RNA content of EV isolates is different when the cells are cultured in EV-depleted FBS-containing medium or FBS-free medium. They concluded that FBS-specific RNA is co-isolated with cell-culture-derived extracellular RNA and interferes with downstream RNA analysis [28].

As serum-free medium, Li et al. used Opti-MEM instead of DMEM, which is specifically optimized to maintain cell viability with reduced serum amounts. They investigated the quantity and protein composition of EVs secreted from N2a mouse neuroblastoma cells cultured in 10% EV-depleted FBS-containing DMEM and in serum-free Opti-MEM. In accordance with our results, culturing the cells in serum-free medium reduced the cell viability, altered the morphology and increased the EV release of the cells. Their proteomic analysis showed higher levels of certain vesicular proteins, including G-protein and GTPase/Ras-related proteins, in serum-free conditions. Because these proteins are involved in EV biogenesis, the authors hypothesized that their increased expression might lead to the increased release of EVs [29]. Our data support this finding as the increased sEV release was also coupled with a predicted increase in GTPase activity in the case of the w/o FBS sEV sample. Investigating myeloma-derived microvesicles, Sun et al. also demonstrated elevated EV production, altered protein content and functional effects of the vesicles upon FBS deprivation, i.e., reducing the FBS content of the culture media from 10% to 1% [30]. The consistency of these data suggests that the impact of serum starvation on EV biology is not just a unique or cell-type-specific phenomenon.

Another factor which influences the experimental results of EV studies is the applied isolation protocol. Currently, without a consensus method that allows the recovery of a pure EV subpopulation, there are a series of isolation techniques based on different physical or biochemical properties of the target EV subpopulation [19,20]. One of the most frequently used sEV isolation methods is differential ultracentrifugation. However, there are inter-laboratory differences in its step-by-step protocol, which may impact the purity and yield of isolated vesicles and impede the reproducibility and interpretation of experimental findings [31,32].

Here, we deeply characterized the influence of an additional washing step, i.e., a second round of ultracentrifugation, on the purity, proteome and biological function of melanoma sEVs. We showed that the washing could not reduce the number of detected FBS-derived bovine proteins in the isolates, but it significantly decreased the intensity of bovine proteins, compared to the Ctrl sEV samples isolated via one ultracentrifugation cycle. Pathway analyses demonstrated that the functional activity of the sEV isolates on protein translation may also be modified depending on the number of ultracentrifugation steps.

Cvjetkovic et al. observed that rotor type, g-force and ultracentrifugation time affect exosomal RNA and protein yield and contamination by soluble proteins [31]. Inter-laboratory study by Torres Crigna et al. showed that the resulting EV particle yield in ultracentrifugation-based EV isolation may vary depending on the equipment and operator, as well [32]. Using serum samples, Langevin et al. compared three ultracentrifugation-based sEV isolation protocols; they found differences in the EV yield, ncRNA cargo of sEV isolates and co-isolated RNA carrier proteins [33]. These data, along with our detailed proteomic analysis, highlight the impact of the ultracentrifugation protocol variances on the reliability and reproducibility of sEV experimental data.

It is also important to note that today, there is a trend toward the combination of multiple techniques for EV isolation, which may increase isolation efficacy in terms of both yield and purity. Therefore, the purity of the sEV isolates used in this work cannot be compared to the purity of the isolates prepared using combined methods [34,35,36].

After isolation, sEV samples usually have to be stored before downstream analyses for technical reasons. According to the review paper of Jeyaram and Jay, storing at −80 °C is the most suitable option for preserving isolated EV samples. However, storage can induce changes in EV bioactivity [21]. A recent paper showed a significant impact of storage at −80 °C on EV samples in terms of particle loss, purity reduction and fusion phenomena. Therefore, the authors suggested the usage of fresh, non-archival samples in the majority of cases depending on the downstream analyses and experimental settings [37]. 

Using an HEI-OC1 murine auditory cell line, Kalinec et al. compared the proteome of fresh and stored EV preparations using LC-ESI-MS/MS. Samples were stored for up to 4 months at −20 °C, and the authors found no significant differences between the two sample types [38]. Another study investigated mouse bronchoalveolar lavage fluid (BALF) exosomes purified via differential ultracentrifugation. The authors analyzed the EV preparations immediately or after storage at +4 °C or −80 °C using LC-MS/MS. As storage caused exosome leakage or the dissociation of loosely bound ‘peri-exosomal’ proteins, and they re-pelleted the stored vesicles before mass spectrometry, they found significant differences between the protein content of the EVs [39]. Here, we investigated the proteome of the whole sEV isolate after thawing, and no significant impact of storage was found. We believe that proper storage usually does not interfere with the mass spectrometry measurements; however, experimental findings may vary by experimental approach.

In the present proteomic study, we aimed to deepen our knowledge on the influence of different technical parameters on sEV biology and the reliability of experimental data. Lacking gold-standard EV research methods, our findings contribute to setting up the most appropriate experimental protocols for future studies, and to interpret previous EV research data. This analysis of the B16F1 melanoma sEV isolates may provide potential protein candidates (Appendix A) worth investigating in future studies focusing on tumor or sEV biology. Because the database search showed the origin of the identified proteins, we could distinguish the B16F1 cell-culture-derived and FBS-derived proteins. In general, we found that almost 25% of the proteins were identified as bovine proteins (in spite of using sEV-depleted serum), suggesting substantial contamination by the FBS supplementation, which may affect different biological processes and molecular functions, such as enzyme activities in the sEV target cells. We showed that an additional washing step may enhance the purity of sEV preparations isolated via ultracentrifugation, but it also affects the composition and function of mouse proteins. As FBS-derived sEVs are co-isolated with the cell culture vesicles, serum-free medium is used in several studies [17]. Here, we highlighted that it alters not just cell viability and sEV production, but also the information content, i.e., protein cargo of the sEVs; this may influence, for example, the molecular signaling processes in the target cells. We also demonstrated that sEV isolates can be stored at −80 °C before LC-MS/MS analysis as this method of storage did not affect the identified protein set. 

In summary, there is an urgent need to solve the problem caused by the FBS contamination of cell-culture-derived sEV isolates. As serum starvation strongly influences sEV biology, using sEV-depleted FBS may be a better, but not the optimal solution. The development of synthetic media formulations or specific sEV isolation techniques may provide the best approaches for future studies.

## 5. Conclusions

Using serum-free conditions in in vitro sEV studies can be advantageous for downstream analyses, since the exogenous, FBS-derived sEVs do not interfere with the research findings. However, serum starvation has limitations as it influences sEV biology, including the release, content and biological activity of sEVs in different cell cultures, and not just in melanoma. Regarding sEV isolation, it is evidenced that ultracentrifugation protocol variances have an impact on the reliability and reproducibility of sEV experimental data. Additionally, proper storage usually does not interfere with the mass spectrometry measurements. We believe that our detailed proteomic analysis of sEVs helps in designing future in vitro studies and gives further insights into sEV biology.

The main conclusion of this study is that applying serum-free conditions should be considered in in vitro sEV studies. Developments of FBS-replacing supplements that do not modify cellular physiology—including sEV release and properties—may provide a good solution for future studies.

## Figures and Tables

**Figure 1 life-13-00206-f001:**
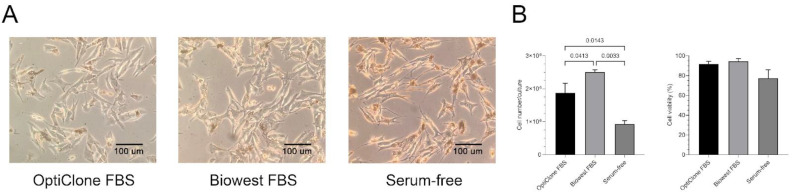
Cell morphology, proliferation and viability under different medium conditions. (**A**) Microscopic images of B16F1 cells cultured in medium containing sEV-depleted OptiClone or Biowest FBS or no FBS (serum-free). (**B**) Diagrams show the cell number and cell viability of the melanoma cultures in the different media. Bars represent mean + SD values (*n* = 3). The *p* values were obtained using ordinary one-way ANOVA followed by Tukey’s post hoc test.

**Figure 2 life-13-00206-f002:**
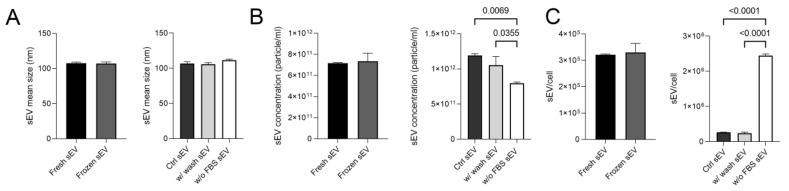
Mean size and concentration of sEVs under different conditions. (**A**) The boxplots show the mean size of the five investigated sEV samples. (**B**) The boxplots show the sEV concentrations measured in the different sEV isolates. (**C**) The boxplots demonstrate the average number of vesicles released per cell. Bars represent mean + SD values (*n* = 3). The *p* values were obtained using Welch’s test and one-way ANOVA followed by Tukey’s post hoc test.

**Figure 3 life-13-00206-f003:**
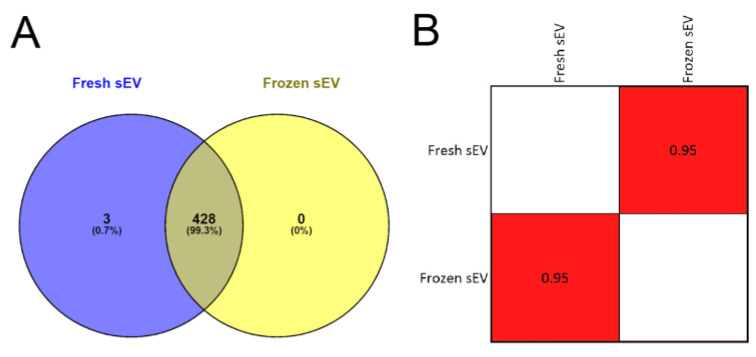
Proteomic analysis of Fresh sEV and Frozen sEV samples. (**A**) Venn diagram illustrates the entire proteome of the Fresh sEV and Frozen sEV samples. (**B**) The correlation matrix shows the comparison of the proteomes of the Fresh sEV and the Frozen sEV samples. Correlation coefficients were obtained using Pearson’s correlation analysis.

**Figure 4 life-13-00206-f004:**
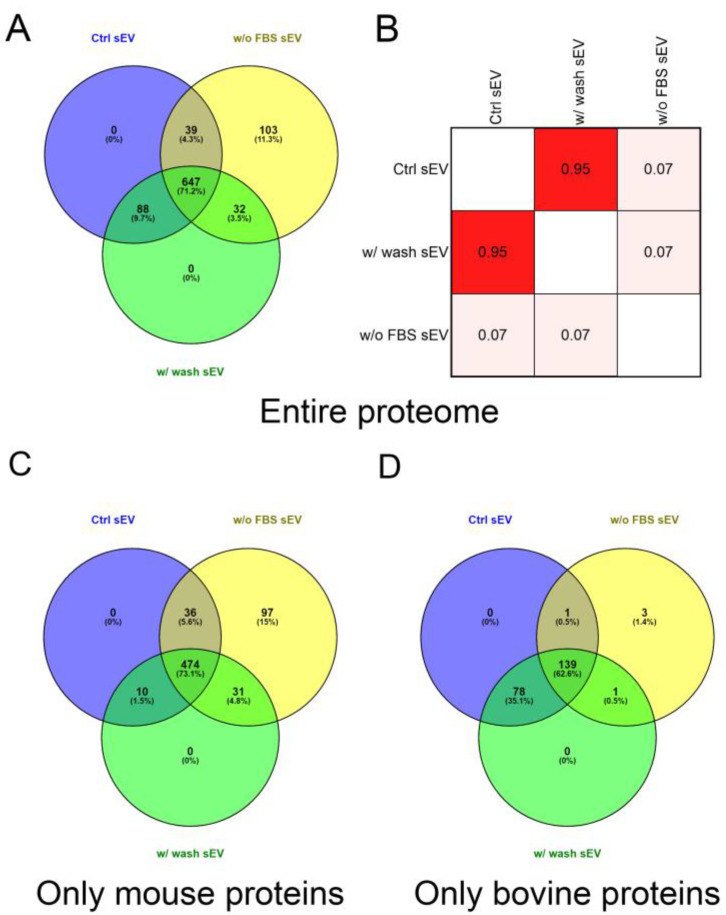
Proteomic analysis and comparison of the Ctrl sEV, w/ wash sEV and w/o FBS sEV samples. (**A**) Venn diagram illustrates the entire proteome of the three samples. (**B**) Correlation matrix shows the comparison of the sEV proteome of the Ctrl sEV, w/ wash sEV and w/o FBS sEV samples. The correlation coefficients were obtained using Pearson’s correlation analysis. Further Venn diagrams illustrate the mouse (**C**) and the bovine (**D**) proteins of the three samples.

**Figure 5 life-13-00206-f005:**
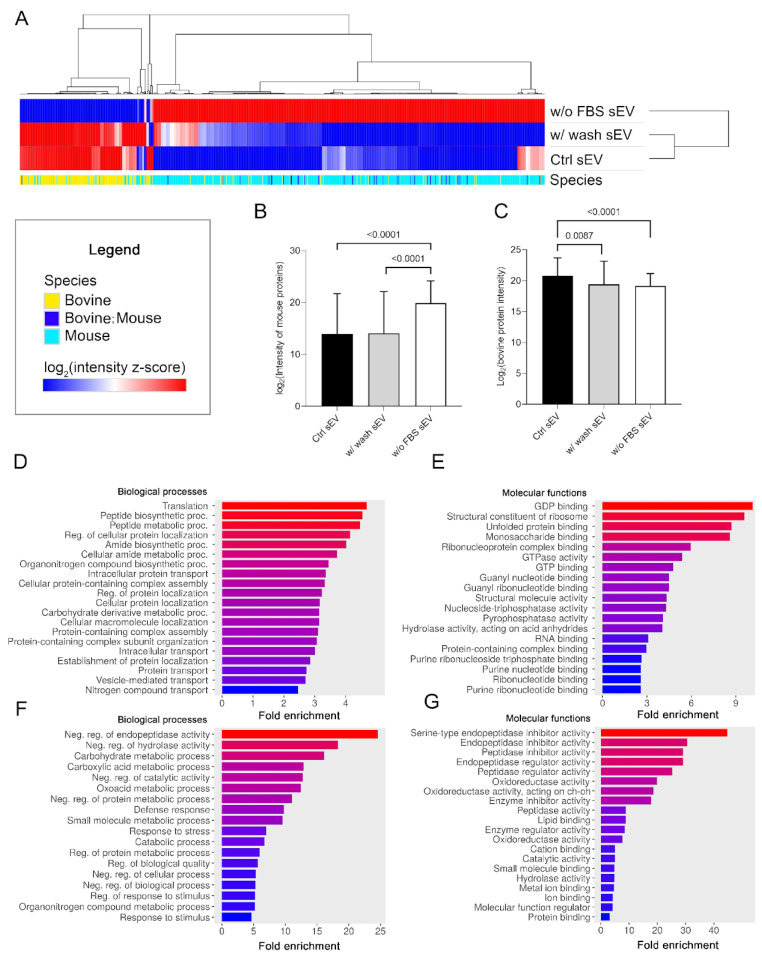
Proteomic pattern and pathway enrichment analysis of the three sEV sample types. (**A**) Hierarchical clustering of the entire proteomic content of the three sEV sample types (columns indicate the proteins). (**B**) The boxplot shows logarithmized intensity of the mouse proteins (mean + SD) of the sEV samples derived from different culturing conditions and isolation procedures. (**C**) The boxplot shows logarithmized intensity of the bovine proteins (mean + SD) of the sEV samples derived from different culturing conditions and isolation procedures. The *p* values were obtained using Welch’s ANOVA followed by Games–Howell post hoc test. Bar charts show the top 20 GO terms for biological processes (**D**) and molecular functions (**E**)—ranked by fold enrichment—following analysis of mouse proteins identified in the Ctrl sEV sample. Bar charts show the top 20 GO terms for biological processes (**F**) and molecular functions (**G**)—ranked by fold enrichment—following analysis of Ctrl sEV-associated bovine proteins. Changing colors of the bars represent the decreasing fold enrichment values.

**Figure 6 life-13-00206-f006:**
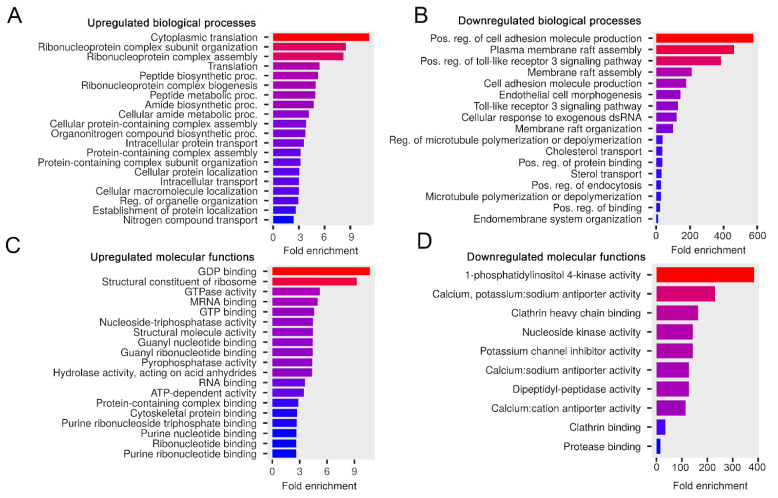
The influence of serum-free conditions on the biological processes and molecular functions of the melanoma sEV proteins. Bar charts show the top 20 GO terms for the elevated (**A**) and downregulated (**B**) biological processes—ranked by fold enrichment—of the associated mouse proteins in the w/o FBS sEV sample compared to the Ctrl sEV sample. Bar charts show the top 20 GO terms for the elevated (**C**) and downregulated (**D**) molecular functions—ranked by fold enrichment—of the associated mouse proteins in the w/o FBS sEV sample compared to the Ctrl sEV sample. Changing colors of the bars represent the decreasing fold enrichment values.

**Table 1 life-13-00206-t001:** Investigated sEV samples.

Terminology	FBS	sEV Isolation	Storage
Fresh sEV	OptiClone	1 × UC	-
Frozen sEV	OptiClone	1 × UC	−80 °C, 3 months
Ctrl sEV	Biowest	1 × UC	-
w/o FBS sEV	-	1 × UC	-
w/ wash sEV	Biowest	2 × UC	-

Abbreviations: FBS—fetal bovine serum; sEV—small extracellular vesicle; UC—ultracentrifugation.

## Data Availability

All datasets generated during the current study are available from the corresponding author upon reasonable request.

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
