# Peer review of "Impact of Experimental Conditions on Extracellular Vesicles’ Proteome: A Comparative Study"

_life, 2023, doi:10.3390/life13010206_

Round 1
Reviewer 1 Report
Gist/Summary: The authors' in their work apply and consider serum-free conditions for inderring in vitro small extra vesicules (sEV) studies. For this they employ, Fetal Bovine Serum (FBS) and conclude that they do not modify cellular physiology whence done with this approach. However, serum starvation has limitations as it influences the sEV. The take home is non-interference of this takien up to the various biologicla activities.
The mansucript is well documented with good figures and statistics employing correlation coefficients, pathway analyses
There must be an emphasis on how this could be the case say withe xosome depleted FBS and otehr scenarios. In othe rwords, the conclusions must delve into future diretcions
The heteroscedasticity is "heterosKedasticity" for regression. Whiel Welch's test was done, the statistical measures and cutoff may be described in the text. Haveings aid this, the q value <0.01 is bit too high for an adjuste dp value. Couldn't it be much better or lesser value so that you could have found more bona fide candidates?
A pictorial methodology will be a nice addition.
All websites/URLs may have last accessed dates
Minor but essential:
FBS may be expanded to Fetal bovine serum. I didn't see it abbreviated.
L 226: pathwayS
L277: referred TO as
L433: ARE decreased
L459: method ( pl remove s)
Reviewer 2 Report
The authors report the effect of FBS, storage of sEV preparations, an additional stage of washing during the isolation procedure, on the proteome of sEV and reproducibility of experimental results.
The topic is interesting. There are some minor and major issues that the authors may want to consider before publication.
Major:
1. As recently published highly purified sEV preparations requires the sequential combinations of different methods (ultracentrifugation, high-resolution density gradient fractionation) [Cell 2019, PMID:30951670; J Extracell Vesicles 2020, PMID: 32944179]. Thus, it should be mentioned that the purity of the sEV preparations in this study is nonoptimal.
2. Please include photos of transmission electron microscopy of sEV preparations. According to independent studies, when sEV are purificated by ultracentrifugation, a large number of proteins, protein complexes, and lipoproteins are co-isolated with sEV [J Extracell Vesicles 2020, PMID: 32944179]. Therefore, it is necessary to control the composition of sEV preparations using transmission electron microscopy and take into account the presence of contaminating structures when analyzing and interpreting the research results.
3. Chapter 3.1: The authors compare the effect of the culture medium on the morphology and proliferation of melanoma cells. were the same number of cells seeded in plates for the comparative study? Please include this information in « 2. Materials and Methods» or « 3.1 Culture medium affected the morphology and proliferation, but not the viability of the cells».
Minor:
Please include w/o FBS sEV, w/ wash sEV in abbreviation or include a more detailed description of these samples at the first mention in the text.
Round 2
Reviewer 2 Report
The authors addressed most of my comments. I recommend it for publication.